# Validation of a hierarchical algorithm to define chronic liver disease and cirrhosis etiology in administrative healthcare data

George Philip[1], Maya Djerboua[2], David Carlone[3], Jennifer A. Flemming[1,2,3,4]*

**1** Translational Institute of Medicine, Queen's University, Kingston, Ontario, Canada, **2** ICES, Queen's University, Kingston, Ontario, Canada, **3** Departments of Medicine, Queen's University, Kingston, Ontario, Canada, **4** Departments of Public Health Sciences, Queen's University, Kingston, Ontario, Canada

* jennifer.flemming2@kingstonhsc.ca

**Data Availability Statement:** The authors are affiliated with ICES and conducted the study in fulfillment of ICES' mandate as a prescribed entity under Ontario's Personal Health Information

## Abstract

### Background and aims

Chronic liver disease (CLD) and cirrhosis are leading causes of death globally with the burden of disease rising significantly over the past several decades. Defining the etiology of liver disease is important for understanding liver disease epidemiology, healthcare planning, and outcomes. The aim of this study was to validate a hierarchical algorithm for CLD and cirrhosis etiology in administrative healthcare data.

### Methods

Consecutive patients with CLD or cirrhosis attending an outpatient hepatology clinic in Ontario, Canada from 05/01/2013–08/31/2013 underwent detailed chart abstraction. Gold standard liver disease etiology was determined by an attending hepatologist as hepatitis C (HCV), hepatitis B (HBV), alcohol-related, non-alcoholic fatty liver disease (NAFLD)/cryptogenic, autoimmune or hemochromatosis. Individual data was linked to routinely collected administrative healthcare data at ICES. Diagnostic accuracy of a hierarchical algorithm incorporating both laboratory and administrative codes to define etiology was evaluated by calculating sensitivity, specificity, positive (PPV) and negative predictive values (NPV), and kappa's agreement.

### Results

442 individuals underwent chart abstraction (median age 53 years, 53% cirrhosis, 45% HCV, 26% NAFLD, 10% alcohol-related). In patients with cirrhosis, the algorithm had adequate sensitivity/PPV (>75%) and excellent specificity/NPV (>90%) for all etiologies. In those without cirrhosis, the algorithm was excellent for all etiologies except for hemochromatosis and autoimmune diseases.

Protection Act. As a result, the authors were authorized, both legally and contractually, to access the data set in a more granular form than approved third party researchers would be permitted to access. The data set that approved third party researchers would be permitted to access would be adjusted to ensure the risk of re-identification of any underlying individuals is low. While data sharing agreements and privacy legislation for the province of Ontario prohibit ICES from making the data set publicly available, access may be granted to those who meet pre-specified criteria for confidential access, available at www.ices.on.ca/DAS. Please see the Methods Section for an outline of all datasets used.

**Funding:** JAF: American Association for the Study of Liver Disease Foundation Clinical, Translational and Outcomes Research Award in Liver Disease (http://www.aasldfoundation.org/); Southeastern Ontario Academic Medical Association New Clinician Scientist Award (https://www.seamo.ca/). The funders has no role in study design, data collection and analysis, decision to publish, or preparation of the manuscript.

**Competing interests:** I have read the journal's policy and the authors of the manuscript have the following competing interests: Jennifer A. Flemming declares investigator-driven research support received from Gilead Sciences Canada, as well as speaker fees from Gilead Sciences, AbbVie, and Lupin Pharmaceuticals. The commercial affiliations of author JAF had no role in the study design, data collection and analysis, decision to publish, or preparation of the manuscript and only provided financial support in the form of author's salaries. This does not alter our adherence to PLOS ONE policies on sharing data and materials.

## Conclusions

A hierarchical algorithm incorporating laboratory and administrative coding can accurately define cirrhosis etiology in routinely collected healthcare data. These results should facilitate health services research in this growing patient population.

## Introduction

Chronic liver disease (CLD) and cirrhosis are the 12th leading causes of death globally [1] and over the past two decades, both the incidence of and mortality from cirrhosis have steadily increased in North America. [2–4] The majority of causes of CLD and cirrhosis have their own distinct epidemiology and natural history with treatment recommendations and outcomes being influenced largely by the underlying etiology. Therefore, the ability to define the cause of CLD and cirrhosis is important both in clinical practice and for clinical research.

The most common causes of CLD and cirrhosis in North America are due to chronic viral hepatitis B (HBV), C (HCV), alcohol-related disease and non-alcoholic fatty liver disease (NAFLD). Together, these conditions are present in approximately 80% of individuals with CLD.[5] More rare causes include autoimmune liver diseases such as autoimmune hepatitis (AIH), primary biliary cholangitis (PBC), and primary sclerosing cholangitis (PSC) and genetic conditions such as hereditary hemochromatosis, Wilson disease, and alpha-1 antitrypsin deficiency. In clinical practice, defining the etiology of CLD and cirrhosis requires careful incorporation of information obtained both from the clinical history and physical examination in addition to results from laboratory, imaging and histologic investigations.[5]

The use of population level administrative healthcare data has evolved as a powerful tool for health services and outcomes research. One of the fundamentals in the use of administrative data is to understand the ability to accurately define specific disease conditions, interventions and outcomes within the databases being used. Given that administrative data does not routinely include details of patients' clinical history, physical exam findings or results of clinical tests, investigators often use surrogates such as physician diagnostic billing codes to identify conditions of interest. However, it is essential to understand if such methods have adequate diagnostic accuracy to identify the specific condition or outcome under investigation. To date, there have been a small number of studies that have validated several CLD and cirrhosis etiologies within administrative healthcare data.[6–10] However, to our knowledge no previous work has evaluated a hierarchical algorithm to define etiology in a defined population with CLD or cirrhosis similar to how etiology is defined in clinical practice.

The aim of this study was to validate a hierarchical algorithm for CLD and cirrhosis etiology incorporating both laboratory and administrative coding in routinely collected healthcare data.

## Methods

### Primary chart abstraction

All consecutive patients with chronic liver disease (elevated AST/ALT > 6 months) or cirrhosis who attended the Kingston Health Sciences Center (KHSC) Liver Clinic between May 1 – August 31, 2013 underwent detailed chart abstraction by a single abstractor (DC). The KHSC Liver Clinic is attended by two subspecialty trained Hepatologists who see a wide variety of

chronic liver conditions referred from the surrounding region of Kingston, Ontario, Canada with a catchment area of approximately 1 million. The majority of patients are referred from primary care practitioners as there are no local community practicing gastroenterologists or hepatologists and KHSC does not perform liver transplantation.

Patient data extracted from the clinic's electronic medical records included patient demographics, laboratory data (including Model for End-stage Liver Disease [MELD], liver enzymes, platelet count) imaging data, endoscopic reports, pathology data, non-invasive fibrosis assessment test results, and any hepatic decompensation events. Cirrhosis was identified based on the presence of any decompensation event (ascites, bleeding varices, encephalopathy, or explicit mention of decompensated cirrhosis) or explicit mention of cirrhosis, or non-bleeding varices. In addition, a liver biopsy result of F4 fibrosis, a non-invasive test result consistent with F4 fibrosis (either serum tests or transient elastography), or imaging consistent with portal hypertension in an individual with known chronic liver disease were also considered diagnostic of cirrhosis. A 5% random sample of charts was re-abstracted by a hepatologist (JAF). Agreement beyond chance on the outcome ascertainment by both abstracters was measured using Cohen's kappa.

## Gold standard: Liver disease etiology

A most responsible cause of liver disease was assigned to each patient based on the overall assessment by the attending Hepatologist evaluating the patient as either HBV, HCV, alcohol-related disease, autoimmune disease (composite of AIH, PBC, PSC), hereditary hemochromatosis, or NAFLD/cryptogenic. NAFLD and cryptogenic were grouped together as the natural history of these two conditions are similar.[11] In cases of viral hepatitis where alcohol was also a contributing factor, the cause of liver disease was assigned as viral hepatitis if the patient remained viremic. In those patients where several causes of liver disease were identified, the one assessed by the Hepatologist as the most likely contributing diagnosis was assigned.

## Administrative databases used for liver disease etiology validation

The validation of liver disease etiology was performed by individual linkage of all abstracted patient data to the routinely collected administrative health care data from the province of Ontario, Canada housed at ICES. ICES is an independent, non-profit research institute funded by an annual grant from the Ontario Ministry of Health and Long-Term Care (MOHLTC). As a prescribed entity under Ontario's privacy legislation, ICES is authorized to collect and use routinely collected health care data for the purposes of health system analysis, evaluation and decision support. Secure access to these data is governed by policies and procedures that are approved by the Information and Privacy Commissioner of Ontario. Ontario provides universal health care coverage for its population of approximately 14 million through the Ontario Health Insurance Program (OHIP). The primary databases used in this analysis were: 1) the Registered Persons Database (RPDB) which includes demographic and vital status information for individuals covered under OHIP; 2) the Canadian Institute for Health Information Discharge Abstract Database (CIHI DAD) which captures diagnostic and procedural information from inpatient hospital admissions; 3) the National Ambulatory Care Reporting System (NACRS) which captures diagnostic and procedural information from ambulatory care and emergency room visits; 4) the OHIP Physician Claims Database which includes all claims made by physicians for universally insured services; 5) the Ontario Laboratory Information System (OLIS) which includes over 90% of all bloodwork results performed by hospitals and clinical laboratories in Ontario from 2007–2015 and; 6) Public Health Ontario (PHO) HBV

and HCV test results from 1997–2015 which processes over 95% of all viral hepatitis testing in the province. These databases were linked using anonymized unique encoded identifiers at the individual level and analyzed at ICES. Patient income quintile and rurality were derived from RPDB and based on area-level demographics of the patient's postal code.[12] Previous hospitalizations and emergency room (ER) visits were determined from CIHI DAD and NACRS. Due to an ICES privacy agreement, data containing small cells (n≤5) are not reportable due to re-identification risk. This study was approved by the Health Sciences Research Ethics Board at Queen's University (DMED 1651–13).

## Administrative algorithm to identify liver disease etiology

To identify causes of chronic liver disease in administrative data from ICES, a hierarchical algorithm was developed adapted from an algorithm previously used in the Veteran's Affairs (VA) administrative data.[13] The algorithm is based on hierarchical criteria similar to how liver disease etiology is assigned in clinical practice that categorizes patients into specific underlying causes of liver disease under the condition that more plausible causes have been excluded (Fig 1). First, patients were assessed for the presence of chronic viral hepatitis through the use of laboratory test results from the PHO Laboratory Information System or OLIS. The presence of a positive HCV RNA or HCV genotype classified an individual as HCV. A positive HCV antibody in isolation was not considered diagnostic for HCV. If negative, a positive HBV DNA or HBV surface antigen was required to define the etiology as HBV. If all viral testing was negative, ICD and OHIP coding from CIHI DAD and NACRS were evaluated for the presence of autoimmune conditions or hereditary hemochromatosis (Table 1) prior to the clinic visit date. Diagnoses in both databases are based on codes from the International Classification of Diseases, 9th (ICD-9, 1988–2001) and 10th revisions (ICD-10, 2002-onwards). If present in the administrative data, they were assigned as either autoimmune or hereditary hemochromatosis. If negative, databases were searched for codes associated with alcohol-related conditions previously used in ICES data holdings (Table 1). [14,15] If all above were negative, then the patient was assigned as having NAFLD/cryptogenic liver disease etiology. The gold standard liver disease etiology from chart abstracted data was then compared to the liver disease etiology diagnosis based on the algorithm.

## Statistical analysis

Descriptive characteristics of the liver disease cohort from KHSC overall and stratified by cirrhosis status were described. Frequencies and proportions were calculated for categorical variables (sex, income quintile, rurality, and patients with previous hospitalization or ED visits) while median and interquartile ranges were calculated for the numeric variables (age, MELD, laboratory values). Differences based on cirrhosis status were compared using t-tests and chi-squared tests.

The algorithm's ability to accurately identify the most plausible cause of CLD and cirrhosis when applied to administrative data in comparison to clinical data was performed stratified by cirrhosis status. Sensitivity with 95% confidence intervals were calculated as the proportion of patients with a specified cause of CLD or cirrhosis identified through administrative data over the number of patients assigned the same cause based on gold standard clinical diagnoses. Specificity with 95% confidence intervals were calculated as the proportion of patients without the specified cause of CLD or cirrhosis in the administrative data over the number of patients without the same cause based on gold standard clinical diagnosis. Positive predicted values (PPV), negative predictive values (NPV), and kappa's agreement with 95% confidence intervals were also calculated where a kappa >0.60 indicates substantial

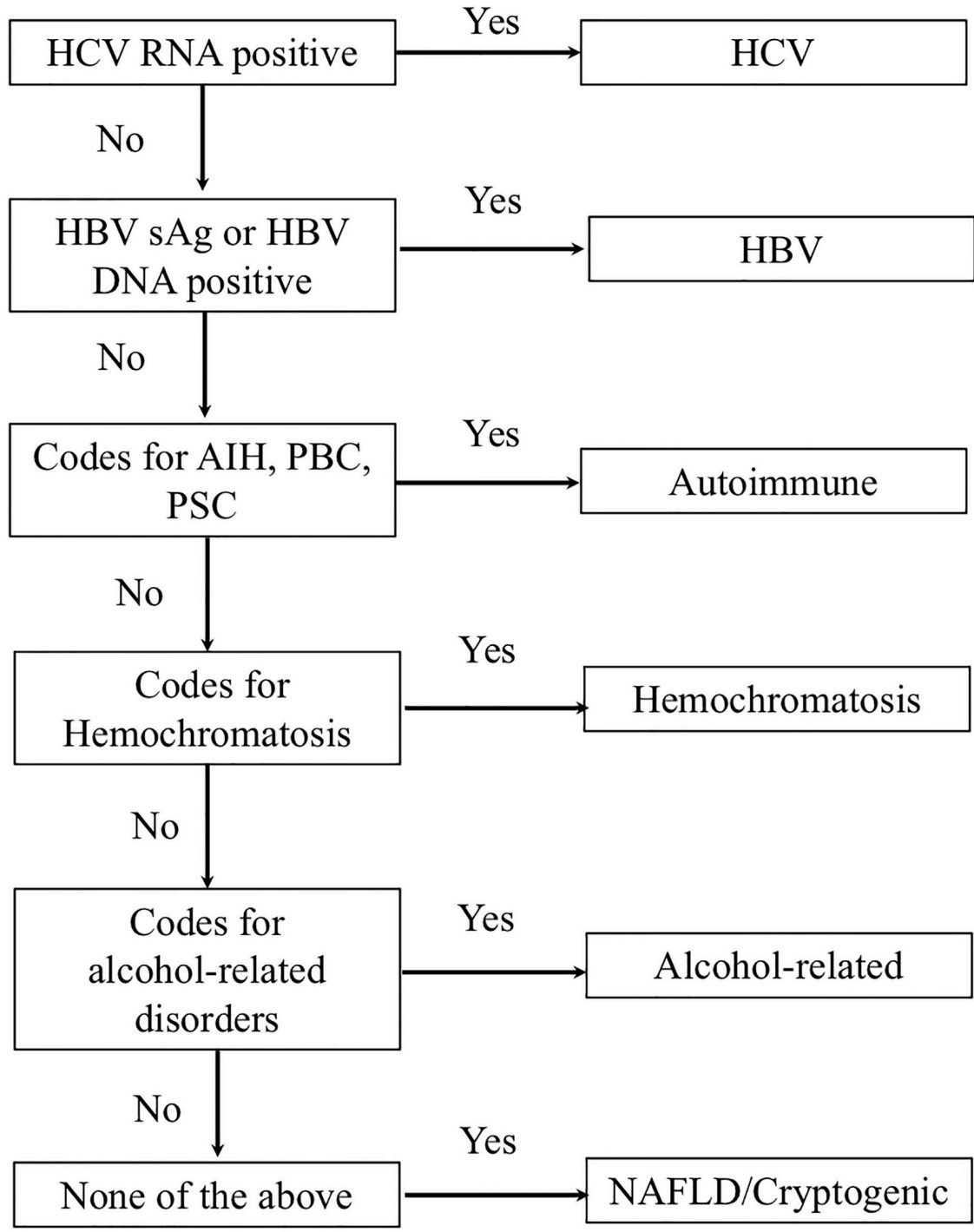

**Fig 1. Hierarchical algorithm to define CLD and cirrhosis etiology in administrative healthcare data.** HCV: hepatitis C; HBV: hepatitis B; sAg: surface antigen; AIH: autoimmune hepatitis; PBC: primary biliary cholangitis; PSC: primary sclerosing cholangitis.

**Table 1. ICD-9 and 10 codes used to define CLD and cirrhosis etiology in ICES data.**

| | ICD-9 Codes | ICD-10 Codes |
|---|---|---|
| **Autoimmune Disease** | | |
| Primary biliary cholangitis or biliary cirrhosis | 571.6 | K74.3 |
| Primary sclerosing cholangitis (PSC) | 576.1 | K83.0 |
| Autoimmune hepatitis (AIH) | 571.42 | K75.4 |
| **Metabolic Disorder** | | |
| Hereditary Hemochromatosis (HH) | 275.0 | E83.10 |
| **Alcohol-related codes** | | |
| Acute intoxication | 305.0 | F100 |
| Harmful alcohol use | 305.0 | F101 |
| Alcohol dependence | 303 | F102 |
| Alcohol withdrawal | 291.0 | F103, F104 |
| Other alcohol-related psychoses | 291 | F105-F109 |
| Accidental or intentional poisoning by alcohol | | X45, Y15, X65 |
| Alcoholic fatty liver | 571.0 | K700 |
| Alcoholic hepatitis | 571.1 | K701 |
| Alcoholic fibrosis and sclerosis of liver | 571.2 | K702 |
| Alcoholic cirrhosis | 571.2 | K703 |
| Alcoholic hepatic failure | | K704 |
| Alcoholic liver disease, unspecified | 571.3 | K709 |
| Alcoholic gastritis | 535.3 | K292 |
| Degeneration of nervous system due to alcohol | | G312 |
| Alcoholic polyneuropathy | 357.5 | G621 |
| Alcoholic myopathy | | G721 |
| Alcoholic cardiomyopathy | 425.5 | I426 |
| Alcohol-induced pancreatitis | | K852, K860 |
| Alcohol-induced pseudo Cushing's syndrome | | E244 |
| Toxic effect of alcohol | 980.0, 980.9 | T510, T519 |
| Finding of alcohol in blood | 790.3 | R780 |
| Maternal care for (suspected) damage to fetus from alcohol | 760.71 | O354, Q860, P043 |

agreement and a kappa >0.80 indicates almost perfect agreement.[16] All analyses were performed using SAS version 9.4 (SAS Institute, Cary, NC, USA).

## Results

### Description of the KHSC cohort

A total of 442 unique patients underwent detailed chart abstraction (Table 2). The median age at the time of the clinic visit was 57 years (IQR 49–62), 261 (59%) were male and 233 (53%) had cirrhosis. Fibrosis assessment was based on non-invasive or clinical decompensation in 336/442 (76%) with the remaining 106/442 (24%) based on liver biopsy. Of those with cirrhosis, 93 (40%) had a history of decompensation. The most common gold standard chronic liver disease etiologies were HCV (199, 45%), followed by NAFLD/cryptogenic (115, 26%), and alcohol-related disease (45, 10%). As expected, those with cirrhosis were older (median age 59 vs. 52 years), had higher MELD scores and lower platelet counts compared to those without cirrhosis. The majority of patients had at least one hospitalization (311, 70%) or ER visit (415, 94%) identified prior to their clinic visit. The Cohen's kappa for the re-abstracted charts (n = 20) showed complete agreement (kappa = 1).

**Table 2. Demographics of patients evaluated in the liver clinic at Kingston Health Sciences Centre May 2013–August 2013.**

| | Overall N = 442 | Cirrhosis N = 233 | No Cirrhosis N = 209 | P-value^ |
|---|---|---|---|---|
| **Age, median (IQR)** | 57 (49–62) | 59 (54–64) | 52 (41–60) | < .001 |
| **Sex, n (%)** | | | | |
| • **Male** | 261 (59) | 150 (64) | 111 (53) | .039 |
| • **Female** | 181 (41) | 83 (36) | 98 (47) | |
| **Income Quintile\*, n (%)** | | | | |
| • **1** | 120 (27) | 64 (28) | 56 (27) | .516 |
| • **2** | 101 (23) | 58 (25) | 43 (21) | |
| • **3** | 82 (19) | 46 (20) | 36 (17) | |
| • **4** | 74 (17) | 33 (14) | 41 (20) | |
| • **5/Missing** | 65 (15) | 32 (14) | 33 (16) | |
| **Rural, n (%)** | 115 (26) | 57 (25) | 58 (28) | .478 |
| **Gold Standard etiology, n (%)** | | | | |
| • **Hepatitis C** | 199 (45) | 115 (49) | 84 (40) | < .001 |
| • **Hepatitis B** | 37 (9) | 13 (6) | 24 (11) | |
| • **Alcohol-related** | 45 (10) | ≤40 (<20) | ≤10 (<5) | |
| • **NAFLD/Cryptogenic** | 115 (26) | 40 (17) | 75 (36) | |
| • **Autoimmune**[†] | 40 (9) | 24 (10) | 16 (8) | |
| • **Hemochromatosis** | 6 (1) | ≤5 (<3) | ≤10 (<5) | |
| **Most recent laboratory values** | | | | |
| **Bilirubin (umol/L), median (IQR)** | 15 (11–22) | 18 (13–30) | 12 (10–16) | < .001 |
| **AST (U/L), median (IQR)** | 40 (28–65) | 50 (34–81) | 32 (24–46) | < .001 |
| **ALT (U/L), median (IQR)** | 36 (23–65) | 36 (23–70) | 36 (22–59) | .245 |
| **Alk-P (U/L), median (IQR)** | 82 (65–117) | 98 (76–136) | 72 (58–89) | < .001 |
| **Albumin (g/L), median (IQR)** | 38 (33–41) | 35 (31–39) | 40 (37–42) | < .001 |
| **INR, median (IQR)** | 1.10 (1.00–1.20) | 1.20 (1.10–1.30) | 1.00 (1.00–1.10) | < .001 |
| **MELD** | 9 (7–12) | 10 (8–12) | 7 (6–8) | < .001 |
| **Platelets ($10^9$), median (IQR)** | 161 (106–220) | 119 (86–166) | 212 (175–261) | < .001 |
| **History of Decompensation, n (%)** | 93 (40) | 93 (40) | n/a | n/a |
| **At least 1 hospitalization before clinic visit, n (%)** | 311 (70) | 176 (76) | 64.6 (135) | 0.012 |
| **At least 1 hospitalization within 1 year before clinic visit, n (%)** | 85 (19) | 62 (27) | 23 (11) | < .001 |
| **At least 1 ER visit before clinic visit, n (%)** | 415 (94) | 223 (96) | 192 (92) | 0.092 |
| **At least 1 ER visit within 1 year before clinic visit, n (%)** | 189 (43) | 120 (52) | 69 (33) | < .001 |

^: comparing those with and without cirrhosis;

\* 1 = lowest, 5 = highest;

[†] includes autoimmune hepatitis, primary biliary cholangitis, primary sclerosing cholangitis

IQR: interquartile range; NAFLD: non-alcoholic fatty liver disease; MELD: model for end-stage liver disease; ER: emergency room.

### Validation of chronic liver disease etiology algorithm

The results of the validation for those with cirrhosis are shown in Table 3. Due to low numbers of patients with both hemochromatosis and cirrhosis, this etiology was not validated. In the patients with cirrhosis, the coding algorithm showed excellent specificity and NPV with values being > 95% for all gold standard diagnoses. Sensitivity and PPV were > 80% for all etiologies with the exception of NAFLD/cryptogenic where the sensitivity was slightly lower at 75% and PPV of 77%. Further, the kappa value was >0.7 for etiologies showing excellent agreement. Results for those without cirrhosis are shown in Table 4. Again, excellent sensitivity and specificity were demonstrated for viral hepatitis, alcohol-related disease, and NAFLD/cryptogenic

**Table 3. Validation of etiology in patients with cirrhosis.**

|  | Sensitivity (95% CI) | Specificity (95% CI) | PPV (95% CI) | NPV (95% CI) | Kappa (95% CI) |
|---|---|---|---|---|---|
| **Hepatitis C** | 0.97 (0.93–0.99) | 1.00 (0.97–1.00) | 1.00 (0.97–1.00) | 0.98 (0.93–0.99) | 0.98 (0.95–1.00) |
| **Hepatitis B** | 0.92 (0.64–1.00) | 0.99 (0.97–1.00) | 0.86 (0.57–0.98) | 1.00 (0.97–1.00) | 0.88 (0.75–1.00) |
| **Alcohol-related** | 0.90 (0.76–0.97) | 0.96 (0.92–0.98) | 0.82 (0.67–0.92) | 0.98 (0.95–0.99) | 0.83 (0.73–0.92) |
| **NAFLD/Cryptogenic** | 0.75 (0.59–0.87) | 0.95 (0.91–0.98) | 0.77 (0.61–0.89) | 0.95 (0.91–0.97) | 0.71 (0.59–0.83) |
| **Autoimmune** | 0.83 (0.63–0.95) | 0.99 (0.97–1.00) | 0.91 (0.71–0.99) | 0.98 (0.95–0.99) | 0.86 (0.73–0.92) |

NAFLD: non-alcoholic fatty liver disease; PPV: positive predictive value; NPV: negative predictive value

**Table 4. Validation of etiology in patients without cirrhosis.**

|  | Sensitivity (95% CI) | Specificity (95% CI) | PPV (95% CI) | NPV (95% CI) | Kappa (95% CI) |
|---|---|---|---|---|---|
| **Hepatitis C** | 0.90 (0.82–0.96) | 1.00 (0.97–1.00) | 1.00 (0.95–1.00) | 0.94(0.88–0.97) | 0.92 (0.86–0.97) |
| **Hepatitis B** | 0.96 (0.79–1.00) | 0.99 (0.97–1.00) | 0.96 (0.79–1.00) | 0.99 (0.97–1.00) | 0.95 (0.89–1.00) |
| **Alcohol-related** | 0.80 (0.28–0.99) | 0.97 (0.94–0.99) | 0.40 (0.12–0.74) | 1.00 (0.97–1.00) | 0.52 (0.21–0.83) |
| **NAFLD/Cryptogenic** | 0.92 (0.83–0.97) | 0.87 (0.80–0.92) | 0.80 (0.70–0.88) | 0.95 (0.90–0.98) | 0.77 (0.68–0.86) |
| **Autoimmune** | 0.56 (0.30–0.80) | 0.99 (0.97–1.00) | 0.90 (0.56–1.00) | 0.96 (0.93–0.99) | 0.67 (0.46–0.88) |
| **Hemochromatosis** | 0.40 (0.05–0.85) | 1.00 (0.97–0.99) | 0.67 (0.09–0.99) | 0.99 (0.96–1.00) | 0.49 (0.06–0.92) |

NAFLD: non-alcoholic fatty liver disease; PPV: positive predictive value; NPV: negative predictive value

etiologies. Although the specificity and NPV for autoimmune liver disease and hemochromatosis were high, the sensitivities were low at 56% and 40% respectively with kappa values of 0.67 and 0.49 respectively.

## Discussion

In this study, we validated a sensitive and highly specific hierarchical algorithm within administrative data to assign a chronic liver disease etiology in patients with cirrhosis. Additionally, to our knowledge, this is the first validation of a hierarchical algorithm to define liver disease etiology within administrative data using a combination of viral hepatitis serology and ICD-9 and 10 codes. The results of this study will facilitate investigators ability to perform both epidemiologic and health services research in patients with cirrhosis using routinely collected administrative healthcare data.

The natural history of CLD and cirrhosis is closely linked to the underlying disease etiology. For example, effective and well tolerated treatments are available for both HCV and HBV that have been shown to alter its natural history. In patients with alcohol-related liver disease, alcohol abstinence is the mainstay of therapy. This is in contrast to NAFLD where, other than lifestyle interventions, no medical therapy is currently approved that has been shown to alter the disease trajectory. Further, from a public health perspective, strategies to identify and manage certain subtypes of patients with liver disease is important. Therefore, it is essential to be able to categorize patients with liver disease into their disease etiology to better understand trends in disease epidemiology, healthcare utilization and clinical outcomes.

Our validation cohort is reflective of a population of patients with chronic liver disease that would be evaluated in a general outpatient Internal Medicine, Gastroenterology, or Hepatology practice. This cohort included men and women both with and without cirrhosis, and also with a history of decompensated liver disease residing both in urban and rural settings. The

most common causes of liver disease in our cohort are reflective of the causes of liver disease in the general population of North America with the majority having either HCV, NAFLD or alcohol-related disease. Therefore, this study has external validity.

The diagnostic accuracy of the algorithm to define liver disease etiology was superior in patients who had cirrhosis. This may be due to a more thorough evaluation in someone with cirrhosis compared to someone without. Alternatively, it may be explained by patients with cirrhosis having more contact with the healthcare system and therefore more diagnostic codes recorded in the medical record. This is reflected by the fact that more patients with cirrhosis were hospitalized or had an ER visit prior to the date of their KHSC clinic visit compared to those without.

Based on previous work which has evaluated study power in diagnostic tests, our cohort of patients from KHSC had adequate power (>80%) to determine the sensitives and specificities for HCV, HBV, alcohol-related, NAFLD/cryptogenic, and autoimmune etiologies however, power was lacking to define hereditary hemochromatosis.[17] The highest diagnostic accuracy was seen in patients with viral hepatitis both with and without cirrhosis. This is likely due to the use of PHO viral serology data used to define these diagnoses as compared to the others which relied on the use of ICD or OHIP coding. Additionally, we were able to define alcohol-related etiology with excellent accuracy especially in those with cirrhosis. These results are very comparable to the validation of both viral hepatitis and alcohol-related disease done in patients with cirrhosis in the VA system in the United States.[10] Overall, the sensitivity and PPV to define patients with NAFLD/cryptogenic cirrhosis was slightly lower than that of viral hepatitis or alcohol-related disease however the accuracy remained acceptable for research purposes. In general, using an algorithm with a high specificity is preferable to identify a cohort of patients as it maximizes the likelihood that individuals have the condition of interest. Further, our results are comparable to the only other validation study of NAFLD in administrative data which was also performed in the VA.[9] Approximately 10% of our cohort had autoimmune liver diseases or hereditary hemochromatosis and therefore, especially in patients without cirrhosis, the accuracy was lower compared to other etiologies. This is further explained by the fact that outpatient OHIP billing codes do not have specific diagnostic codes for the autoimmune liver conditions or hereditary hemochromatosis. Therefore, to receive a specific ICD code for these conditions, the individual would need to have had it recorded during a hospital admission or emergency room visit which is less frequent in those without cirrhosis (Table 1). Previous work has also shown the ability to define PSC in administrative data is suboptimal.[7] Therefore, in patients without cirrhosis, the ability of administrative data to define these etiologies should be taken into consideration.

The results of this study should be considered in light of methodologic limitations. The cohort used for the validation was derived from a single center outpatient Hepatology practice and therefore may not be reflective of the entire population of patients with CLD and cirrhosis. However, we believe this algorithm would be applicable to any individual assessed for CLD etiology who has received a general evaluation for causes of CLD as recommended by guidelines given that our cohort was derived largely by referrals from primary care practitioners. Additionally, the catchment of KHSC is approximately 1 million individuals in Ontario with 25% rural residence, however, the ethnic diversity of our catchment area would be less than that of a major urban center. Secondly, we did not evaluate the ability of the administrative data to identify two or more causes of chronic liver disease. Third, due to low numbers of patients with autoimmune liver conditions, we were unable to evaluate each specific condition (AIH, PBC, PSC) separately. Further, there were no patients in the cohort who had diagnoses of alpha-1 antitrypsin deficiency or Wilson disease and therefore the ability of administrative data to define these chronic liver diseases is unknown. Further, in large administrative data, patients with these diagnoses

using this algorithm would be grouped into the NAFLD/cryptogenic category. However, given the rarity of these conditions, it would only contribute to < 1% of the CLD/cirrhosis population. Finally, this validation was done in administrative data from a universally insured healthcare system and may not be generalizable to other types of administrative health data.

In conclusion, the use of a hierarchical coding algorithm in administrative healthcare data is able to define CLD and cirrhosis etiology with excellent diagnostic accuracy using a combination of viral hepatitis serology and administrate diagnostic coding, especially in individuals with cirrhosis. These results should facilitate future health services research in this growing patient population.

## Acknowledgments

This study was supported by ICES, which is funded by an annual grant from the Ontario Ministry of Health and Long-Term Care (MOHLTC). The opinions, results, and conclusions reported in this paper are those of the authors and are independent from the funding sources. No endorsement by ICES or the MOHLTC is intended or should be inferred. Parts of this material are based on data and information compiled and provided by CIHI, MOHLTC, and PHO. However, the analyses, conclusions, opinions, and statements expressed herein are those of the author, and not necessarily those of CIHI, MOHLTC, and PHO.

## Author Contributions

**Conceptualization:** George Philip, Jennifer A. Flemming.

**Data curation:** David Carlone, Jennifer A. Flemming.

**Formal analysis:** Maya Djerboua.

**Funding acquisition:** Jennifer A. Flemming.

**Methodology:** George Philip, Maya Djerboua, Jennifer A. Flemming.

**Project administration:** Jennifer A. Flemming.

**Supervision:** Jennifer A. Flemming.

**Validation:** Maya Djerboua.

**Writing – original draft:** George Philip.

**Writing – review & editing:** Maya Djerboua, David Carlone, Jennifer A. Flemming.

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
