## [Decision Letter · Decision Letter 0]

14 Nov 2019

PONE-D-19-27866

Validation of a hierarchical algorithm to define chronic liver disease and cirrhosis etiology in administrative healthcare data

PLOS ONE

Dear Dr. Flemming,

Thank you for submitting your manuscript to PLOS ONE. After careful consideration, we feel that it has merit but does not fully meet PLOS ONE’s publication criteria as it currently stands. Therefore, we invite you to submit a revised version of the manuscript that addresses the points raised by two reviewers during the review process.

We would appreciate receiving your revised manuscript by Dec 29 2019 11:59PM. To enhance the reproducibility of your results, we recommend that if applicable you deposit your laboratory protocols in protocols.io, where a protocol can be assigned its own identifier (DOI) such that it can be cited independently in the future. For instructions see: http://journals.plos.org/plosone/s/submission-guidelines#loc-laboratory-protocols

We look forward to receiving your revised manuscript.

Kind regards,

Wenyu Lin, PhD

Academic Editor

PLOS ONE

Journal Requirements:

2. In the ethics statement in the manuscript and in the online submission form, please provide additional information about the patient records used in your retrospective study, including: a) whether all data were fully anonymized before you accessed them and b) the date range (month and year) during which patients' medical records were accessed,

Reviewers' comments:

Reviewer's Responses to Questions

**Comments to the Author**

1. Is the manuscript technically sound, and do the data support the conclusions?

Reviewer #1: Yes

Reviewer #2: Yes

2. Has the statistical analysis been performed appropriately and rigorously? 

Reviewer #1: Yes

Reviewer #2: Yes

3. Have the authors made all data underlying the findings in their manuscript fully available?

Reviewer #1: Yes

Reviewer #2: Yes

4. Is the manuscript presented in an intelligible fashion and written in standard English?

Reviewer #1: Yes

Reviewer #2: Yes

5. Review Comments to the Author

Reviewer #1: In this study, George Philip et al. validated a hierarchical algorithm for CLD and cirrhosis etiology in administrative healthcare data. Diagnostic accuracy of a hierarchical algorithm incorporating both laboratory and administrative codes to define etiology was evaluated by calculating sensitivity, specificity, positive (PPV) and negative predictive values (NPV), and kappa’s agreement. The use of a hierarchical coding algorithm is able to define CLD and cirrhosis etiology with excellent diagnostic accuracy using a combination of viral hepatitis serology and administrate diagnostic coding, especially in individuals with cirrhosis.

The findings are overall interesting, even the cohort is rather small. I have a number of major issues with this manuscript.

Specific comments to improve the study are listed below:

1. Did the authors perform power calculation? There were 442 patients, is that enough?

2. Cirrhosis was identified based on the presence of any decompensation event. In addition, a liver biopsy result of F4 fibrosis. This is not the golden diagnose. How many patients were diagnosed by these two methods respectively?

3. In the table 3 and 4, the authors should describe specific data, not just Sensitivity, Specificity, PPV etal.

4. In the table 4, “Autoimmune”, the Specificity, PPV and NPV were all high, but only Sensitivity was low? Why?

5. In the table 4, “Hemochromatosis”, the Specificity and NPV were high, but Sensitivity and PPV were low? Please explain the results in discussion. Is it due to the small number of patients? Please offer the clear numbers.

6. there were some limitations in the manuscript, such as all patients from one city, but the authors did not refer some shortages.

Reviewer #2: In the manuscripts, a hierarchical algorithm is applied to find etiology of CLD/cirrhosis by using administrative healthcare data, and the resulting etiology is compared to golden standard identified ones. The data is comprehensive and statistics are valid. There’s some concerns:

• In the abstracts, the term “sensitivity/NPV” and “specificity/PPV” is mismatched.

• In the introduction, author states “The most common causes of CLD and cirrhosis in North America are secondary to chronic viral hepatitis B (HBV)….... ”, so what’s primary causes?

• In Table 2, Golden standard etiology categorical data showed less accurate number and percentage of Alcoholic related and hemochromatosis, could author supply addition explanation?

• “93 (40%) had a history of decompensation” did not appeared in Table 2

• The data did not fully support the conclusion “A hierarchical algorithm incorporating laboratory and administrative coding can accurately define CLD and cirrhosis etiology in routinely collected healthcare data”. For in non-cirrhosis patients, the sensitivities were low at 56% and 40% respectively with kappa values of 0.67 and 0.49 respectively. Author should be more cautious to make the conclusion.

6. PLOS authors have the option to publish the peer review history of their article (what does this mean?). If published, this will include your full peer review and any attached files.

Reviewer #1: No

Reviewer #2: No

---

## [Author Response · Author response to Decision Letter 0]

7 Jan 2020

POINT BY POINT REPONSE TO REVIEWERS

PONE-D-19-27866: Validation of a hierarchical algorithm to define chronic liver disease and cirrhosis etiology in administrative healthcare data

We would like to sincerely thank the editorial board and the peer reviewers for the review and critique of our manuscript with the opportunity to re-submit a revised version. We believe that we have been able to respond to all comments and concerns and feel that this review has greatly improved the manuscript. Please see the below responses and attached revised manuscript.

Reviewer #1: In this study, George Philip et al. validated a hierarchical algorithm for CLD and cirrhosis etiology in administrative healthcare data. Diagnostic accuracy of a hierarchical algorithm incorporating both laboratory and administrative codes to define etiology was evaluated by calculating sensitivity, specificity, positive (PPV) and negative predictive values (NPV), and kappa’s agreement. The use of a hierarchical coding algorithm is able to define CLD and cirrhosis etiology with excellent diagnostic accuracy using a combination of viral hepatitis serology and administrate diagnostic coding, especially in individuals with cirrhosis.

The findings are overall interesting, even the cohort is rather small. I have a number of major issues with this manuscript.

Specific comments to improve the study are listed below:

1. Did the authors perform power calculation? There were 442 patients, is that enough?

Response: Thank you for bringing this to our attention – changes made. You are right to point out that we did not make any comments regarding the study power in the original manuscript. Given that we are using a hierarchical algorithm to define different causes of cirrhosis and CLD our study is a bit different than a standard diagnostic study as we are looking at the power to determine the presence/absence of several different disease states which is dependent on the prevalence of the condition in the group at study. However, we have now calculated the power we had by etiology based on the following manuscript (J Clin Diagn Res. 2016 Oct; 10(10): YE01–YE06). We believe we have adequate power to define the sensitivity and specificity for 1) HCV (based on an anticipated prevalence of 20%-30%, we required between 50-200 patients for sensitivity/specificity >90% [total cases 199]); 2) alcohol-related disease (based on anticipated prevalence of 20%, we required ~25-50 patients for sensitivity/specificity >80% [total cases 45]); 3) NAFLD/cryptogenic (based on anticipated prevalence of 20%, we required ~107 cases for sensitivity/specificity >90% [total cases 115]); HBV (based on anticipated prevalence 5% we required ~5-20 patients for sensitivity/specificity >80% [total cases 37]); autoimmune (based on anticipated prevalence 5% we required ~5-20 patients for sensitivity/specificity >80% [total cases 40]). Our power to define hemochromatosis is not ideal given the low number of cases (total of 6 only) which we had commented on previously. We have clarified this in the discussion with reference to the above-mentioned paper (see Discussion paragraph 5). 

2. Cirrhosis was identified based on the presence of any decompensation event. In addition, a liver biopsy result of F4 fibrosis. This is not the golden diagnose. How many patients were diagnosed by these two methods respectively?

Response: Thank you for this comment – changes made. The definition of cirrhosis is F4 fibrosis on liver biopsy based on the Metvair staging system and is the gold standard to diagnose cirrhosis (J Hepatol. 2007: 47:598-607). However, in clinical practice, an individual with a known chronic liver disease with a liver decompensation event is also considered to have cirrhosis and biopsy is not required. We have clarified in the results section that in the cohort 336/442 (76%) had fibrosis staging based on non-invasive or clinical data and 106/442 (24%) were staged based on biopsy data. See results section paragraph 1. 

3. In the table 3 and 4, the authors should describe specific data, not just Sensitivity, Specificity, PPV etal.

Response: We appreciate this comment however as written it is unclear exactly what specific data the reviewer is referring to. In studies of diagnostic accuracy, it is important to describe sensitivity, specificity, positive predictive value, negative predictive value and the kappa. If there are other parameters that we are missing we are hoping the reviewer could clarify this statement or make a specific suggestion and we are happy to respond to this omission.

4. In the table 4, “Autoimmune”, the Specificity, PPV and NPV were all high, but only Sensitivity was low? Why?

Response: Thank you for this observation. Unlike many of the other chronic liver diseases, the ability to make a diagnosis of autoimmune hepatitis is based on a scoring system of clinical criteria which includes autoimmune markers, liver biopsy findings and the lack of other clinical diagnoses. Given that the sensitivity was only low in those without cirrhosis, we have two potential explanations. First, it is likely that there was a lower frequency of liver biopsies being performed to secure the diagnosis of autoimmune liver disease in individuals without cirrhosis as this is commonly done in clinical practice. Secondly, unlike ICD coding, outpatient OHIP billing codes do not have codes specific to the autoimmune liver diseases (including PBC and PSC). Therefore, in individuals without cirrhosis who are less likely to be admitted to hospital or visit the emergency room, the sensitivity is lower than in those with cirrhosis (sensitivity 0.83). 

5. In the table 4, “Hemochromatosis”, the Specificity and NPV were high, but Sensitivity and PPV were low? Please explain the results in discussion. Is it due to the small number of patients? Please offer the clear numbers.

Response: Thanks – changes made. Typically, in diagnostic accuracy studies, there is always a trade-off between sensitivity and specificity with one going down as the other goes up. Similar to the question #4 response above, outpatient OHIP billing codes are not specific for hemochromatosis, therefore an ICD code for hemochromatosis would only be captured if the individual was admitted to hospital or had an emergency room visit. Table 2 shows that hospitalization/ER visits were less frequent in those without cirrhosis. We have added a comment regarding this in the Discussion section, paragraph 5. Unfortunately, due to privacy agreements we are un able to describe cells with N <=5 as outlined in the methods section. 

6. there were some limitations in the manuscript, such as all patients from one city, but the authors did not refer some shortages.

Response: Thanks – change made. We have an entire paragraph dedicated to the limitations of the study. Please see the limitations section of the Discussion, paragraph 6 where four major limitations are discussed (1: single center study; 2: inability to define two causes of liver disease; 3: low numbers of the rarer liver conditions; 4: databases from universal healthcare system). We have expanded upon the fact that these patients were all from one center as suggested. 

Reviewer #2: In the manuscripts, a hierarchical algorithm is applied to find etiology of CLD/cirrhosis by using administrative healthcare data, and the resulting etiology is compared to golden standard identified ones. The data is comprehensive and statistics are valid. There’s some concerns:

• In the abstracts, the term “sensitivity/NPV” and “specificity/PPV” is mismatched.

Response: Thanks for pointing this out! Change made see abstract.

• In the introduction, author states “The most common causes of CLD and cirrhosis in North America are secondary to chronic viral hepatitis B (HBV)….... ”, so what’s primary causes?

Response: Thanks for pointing out that this sentence was not clear to readers. Please see change to the sentence in the Introduction, paragraph 2.

• In Table 2, Golden standard etiology categorical data showed less accurate number and percentage of Alcoholic related and hemochromatosis, could author supply addition explanation?

Response: Thank you for pointing this out. We believe you are referring to the total number of patients with alcohol-related liver disease and hemochromatosis. Given that this cohort was taken from a group who is being seen in a referral center, it likely reflects that patients with alcohol-related liver disease maybe: 1) less likely to be referred or 2) less likely to attend outpatient appointments compared to patients with other causes of liver disease. With respect to hemochromatosis, typically patients without liver disease may be managed with phlebotomy by colleagues in Hematology and may not present to Hepatology clinics. 

• “93 (40%) had a history of decompensation” did not appeared in Table 2

Response: Thanks - change made. Please see updated Table 2.

• The data did not fully support the conclusion “A hierarchical algorithm incorporating laboratory and administrative coding can accurately define CLD and cirrhosis etiology in routinely collected healthcare data”. For in non-cirrhosis patients, the sensitivities were low at 56% and 40% respectively with kappa values of 0.67 and 0.49 respectively. Author should be more cautious to make the conclusion.

Response: Thank you – change made. Please see revised Abstract and Discussion paragraph 1.

---

## [Decision Letter · Decision Letter 1]

3 Feb 2020

Validation of a hierarchical algorithm to define chronic liver disease and cirrhosis etiology in administrative healthcare data

PONE-D-19-27866R1

Dear Dr. Flemming,

We are pleased to inform you that your manuscript has been judged scientifically suitable for publication and will be formally accepted for publication once it complies with all outstanding technical requirements.

With kind regards,

Wenyu Lin, PhD

Academic Editor

PLOS ONE

Additional Editor Comments (optional):

Reviewers' comments:

Reviewer's Responses to Questions

**Comments to the Author**

1. If the authors have adequately addressed your comments raised in a previous round of review and you feel that this manuscript is now acceptable for publication, you may indicate that here to bypass the “Comments to the Author” section, enter your conflict of interest statement in the “Confidential to Editor” section, and submit your "Accept" recommendation.

Reviewer #1: (No Response)

Reviewer #2: All comments have been addressed

2. Is the manuscript technically sound, and do the data support the conclusions?

Reviewer #1: (No Response)

Reviewer #2: Yes

3. Has the statistical analysis been performed appropriately and rigorously? 

Reviewer #1: (No Response)

Reviewer #2: Yes

4. Have the authors made all data underlying the findings in their manuscript fully available?

Reviewer #1: (No Response)

Reviewer #2: Yes

5. Is the manuscript presented in an intelligible fashion and written in standard English?

Reviewer #1: (No Response)

Reviewer #2: Yes

6. Review Comments to the Author

Reviewer #1: (No Response)

Reviewer #2: (No Response)

7. PLOS authors have the option to publish the peer review history of their article (what does this mean?). If published, this will include your full peer review and any attached files.

Reviewer #1: No

Reviewer #2: No

---

## [Editor Report · Acceptance letter]

6 Feb 2020

PONE-D-19-27866R1 

Validation of a hierarchical algorithm to define chronic liver disease and cirrhosis etiology in administrative healthcare data 

Dear Dr. Flemming:

I am pleased to inform you that your manuscript has been deemed suitable for publication in PLOS ONE. Congratulations! Your manuscript is now with our production department. 

With kind regards,

on behalf of

Dr. Wenyu Lin 

Academic Editor

PLOS ONE